# EWMV: An algorithm to improve the efficiency of conformal methods

## Abstract

Conformal prediction is a framework that augments a machine learning model to return a prediction set in lieu of a single prediction. Although these sets contain the correct answer with a guaranteed probability, their size can be ineffectively large and thus lead to costly erroneous decisions. To mitigate this, we propose EWMV, an algorithm that leverages the available calibration data to aggregate multiple accessible predictors into a single, smaller conformal predictor. Empirical evidence across a variety of tasks and conformal methods suggests EWMV often produces smaller and more efficient prediction sets than any of the individual predictors being aggregated. Accordingly, these findings encourage a new paradigm to improve the efficiency of conformal methods with two readily available resources: calibration data and a plethora of pre-trained predictors.

## 1 Introduction

Accurately quantifying the uncertainty of a machine learning (ML) model's prediction enables the identification, and proper management, of cases the model is likely to be wrong about. This is crucial to mitigate errors in costly decision making pipelines where, for instance, a false positive leads to futile clinical trials (Jin & Candes, 2023) or innocent incarceration (Romano et al., 2020a) and a false negative delays time-critical treatments (Angelopoulos et al., 2024; García et al., 2024). Conformal prediction is an increasingly popular strategy to quantify a model's uncertainty. It does so by mapping an input to a subset of the label space known as the prediction set. The larger the set, the more uncertain the prediction is. To produce accurate uncertainty estimates, conformal methods aim to be "valid" and "efficient". Intuitively, "valid" limits the proportion of times the true answer is not present in the prediction set; and "efficient" corresponds to smaller sets. In steps towards improving efficiency, the conformal model aggregation (CMA) literature has adopted a model selection paradigm and proposed algorithms to identify the most efficient conformal predictor (i.e. smallest expert) from a collection of valid predictors (Gasparin & Ramdas, 2024a; Liang et al., 2024; Yang & Kuchibhotla, 2025). But what if instead we could combine the individual predictors in a way that preserves validity and further improves efficiency? To answer this question, we propose EWMV (Estimated weighted majority vote -Algorithm 1), an aggregation algorithm that leverages the weighted majority vote (WMV) algorithm (Gasparin & Ramdas, 2024b) to combine multiple conformal predictors into a valid and more efficient predictor. EWMV preserves validity (proposition 5.1) and, empirically, we also observe it improves the efficiency of four different conformal methods (i.e. APS (Angelopoulos & Bates, 2022), RAPS (Angelopoulos et al., 2022), TRAQ (Li et al., 2024), CC (García et al., 2024)) when applied to synthetic multiclass classification (Section 6.1), image classification (Section 6.2), natural question answering (Section 6.3) and risk stratification (Section 6.4), respectively. In this paper, we show with extensive testing that EWMV has practical execution times (Section 6.5); we compare EWMV with a variety of heuristics and other conformal model aggregation algorithms (Section 6.6); we delve deeper into the empirical coverage behavior of the aggregated predictor (Section 6.7); and we observe experimentally that performance improves monotonically, on average, the more predictors we aggregate (Section 6.8). Our results show that EWMV can be regarded as a new paradigm to improve the efficiency of conformal methods by leveraging two readily available resources: pre-estimated models and the calibration dataset.

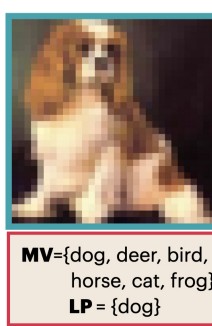 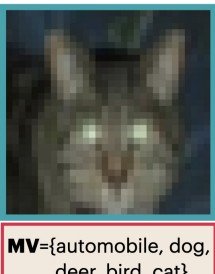 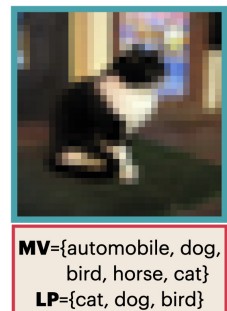

**MV**={dog, deer, bird, horse, cat, frog}
**LP** = {dog}

**MV**={automobile, dog, deer, bird, cat}
**LP**={cat, dog}

**MV**={automobile, dog, bird, horse, cat}
**LP**={cat, dog, bird}

Figure 1: Conformal prediction sets from baseline aggregation method (**MV**) and EWMV (**LP** variant) under corresponding CIFAR10 images.

## 2 RELATED WORKS

The idea of combining conformal prediction sets stems from cross-validation conformal methods (Vovk, 2015; Barber et al., 2021; Angelopoulos et al., 2025). These aim to improve the computational/statistical tradeoff between full-conformal prediction and split-conformal prediction but are not particularly concerned with set size. In the exploration of preserving validity and improving efficiency, the conformal aggregation literature can be broadly categorized into p-value combination methods (Campagner et al., 2024; Vovk & Wang, 2020; Toccaceli & Gammerman, 2019; Toccaceli, 2019; Cherubin, 2019; Toccaceli & Gammerman, 2017; Balasubramanian et al., 2015; Qin et al., 2025) and set combination methods (Gasparin & Ramdas, 2024b;a; Cherubin, 2019; Liang et al., 2024; Yang & Kuchibhotla, 2025).

P-value combination aggregates multiple conformal p-values, for a given label, into a single p-value. This combined p-value is then used to construct the final conformal prediction set. We roughly categorized the methods as follows: quantile methods like Fisher and SNF (Balasubramanian et al., 2015), merging methods like geometric average and arithmetic average (Vovk & Wang, 2020), order statistic methods like min. and max. (Vovk & Wang, 2020), estimation methods like NCA, ECDF, NP-V-Matrix (Balasubramanian et al., 2015; Toccaceli & Gammerman, 2019). Overall the Fisher quantile method is the most frequently recommended (Balasubramanian et al., 2015; Toccaceli & Gammerman, 2017; Toccaceli, 2019). However, a recent study (Campagner et al., 2024) empirically ranks MV (a prediction set combination method) higher in efficiency than the Fisher method and other p-value combination approaches. In our experiments we observe EWMV has superior performance to p-value aggregation approaches. On a similar note, the work of Luo & Zhou; Tawachi & Laufer-Goldshtein (2025) can be characterize as a form of score-level aggregation. Methodologically, this is complimentary to our proposal, as we can use EWMV combine prediction sets that were constructed with score-level aggregation, either by linearly combining multiple scores or by constructing a multidimensional predictor from multiple heads of the same predictor. That said, a key distinction is that, by virtue of doing aggregation post-quantile computation, we are able to provide three tractable optimization formulations for the weight estimation problem, each varying in computational complexity and empirically validated. Accordingly, our post-conformalization aggregation approach scales better than the pre-conformalization brute-force search approach Luo & Zhou.

Prediction set combination aggregates conformal predictors at the set level rather than at the p-value level. These algorithms are some variant of weighted majority vote (WMV) (Cherubin, 2019). In the theoretical exploration of (Gasparin & Ramdas, 2024b), WMV is parametrized by a weight vector that lives in a probability simplex and, for any weight in this simplex, conservative validity is guaranteed. Unfortunately, the chosen weights may negatively affect efficiency (i.e. one of the individual sets is smaller on average), and thus renders WMV useless (See Tables 4, 3 below and Table 2 from Gasparin & Ramdas (2024b)).

The work by Gasparin & Ramdas (2024a) explores weight estimation for the sequential, non-i.i.d. data setting. According to the authors, in the i.i.d. setting, their algorithm effectively selects the single model with the smallest prediction set (akin to finding the "best" expert). However, in the

same setting, our proposal empirically showcases more efficient sets than any of the aggregated models individually, and thus better than the "best expert" model. Expert selection is an active area of research (Liang et al., 2024; Yang & Kuchibhotla, 2025). Notably, the approach by Liang et al. (2024), estimates the smallest conformal set without splitting the calibration data further, nor compromising validity (a limitation of Yang & Kuchibhotla (2025)). This is useful in situations where data is scarce and splitting the calibration set is unreasonable. However, according to our experimental results, granted enough data is available for a split, we can outperform expert selection.

With respect to recent methodological developments in conformal prediction, this work stands as complementary. Rather than posit a new conformal method to guarantee validity or improve efficiency in a new setting (e.g. medical QA (Cherian et al.), class conditional on many classes (Ding et al.), with multiple scores available Luo & Zhou), our work proposes an algorithm to aggregate multiple such conformal predictors. We test our methodology using conformal methods for image classification (APS, RAPS (Angelopoulos et al., 2022)), open-ended question answering (TRAQ (Li et al., 2024)) and risk stratification (CC (Garcıa et al., 2024)) but the scope of the methodology extends beyond and may be used in conjunction with other recent proposals. Lastly, recent work has aimed to optimize efficiency in the context of covariate shifts (Kiyani et al., 2024; Ge et al.). While these work provides a principled way to handle covariates shift, the scalability is limited by the difficulty of the optimization. For instance, the optimization formulation of Kiyani et al. (2024) is a saddle point problem and the proposed gradient descent ascent method may not necessarily converge. This limits its applicability in the i.i.d. setting, where the optimization we formulate can be readily solved with off-the-shelf LP and MILP solvers (Gurobi Optimization, LLC, 2024) and thus are more amenable for practical applications.

## 3 METHODOLOGY

Consider a classification task over a space of features $\mathcal{X}$ and countable classes $\mathcal{Y}$. Suppose we have a sequence of i.i.d. samples $D_n = ((X_1, Y_1), ..., (X_n, Y_n)) \in (\mathcal{X} \times \mathcal{Y})^n$ and let $X_{n+1}$ represent a test feature to classify. Conformal prediction uses the sample $D_n$, a non-conformity score (typically from a pre-estimated probabilistic classifier $f : \mathcal{X} \to \mathcal{Y}$), and a user-specified error level $\alpha \in (0, 1)$, to construct a set-valued classifier (i.e. $C^{(\alpha)} : \mathcal{X} \to 2^{\mathcal{Y}}$). [1] For instance, in Figure 1 we can observe the prediction of two conformal classifiers for a given image. The advantage of using $C^{(\alpha)}$, instead of the underlying model $f$, is that the true label $Y_{n+1}$ is excluded from $C^{(\alpha)}(X_{n+1})$ no more than $\alpha$ proportion of the time. This property is often referred to as validity and is formalized as $\mathbb{P}(Y_{n+1} \notin C^{(\alpha)}(X_{n+1})) \leq \alpha$, where the probability $\mathbb{P}$ is taken w.r.t. the randomness in both the calibration data $D_n$ (used to construct $C^{(\alpha)}$) and the test point $(X_{n+1}, Y_{n+1})$. Define $[M] \coloneqq \{1, ..., M\}$ and let $(C_m^{(\alpha)})_{m \in [M]}$ be a collection of $M$ distinct conformal predictors with error level $\alpha$. The goal of conformal model aggregation is to combine this collection and produce a new conformal predictor $\Gamma^{(\alpha)}$ that preserves validity (i.e. $\mathbb{P}(Y_{n+1} \notin \Gamma^{(\alpha)}(X_{n+1})) \leq \alpha$) and is more efficient, in the sense of producing smaller sets than any individual predictor in the collection. More efficient can be precisely stated as $\forall_{m \in [M]}(\mathbb{E}_X |\Gamma^{(\alpha)}(X)| \leq \mathbb{E}_X |C_m^{(\alpha)}(X)|)$ where $|\cdot|$ measures the cardinality of the predicted set and the expectation $\mathbb{E}_X$ is only w.r.t. $X \sim \mathbb{P}_X$ ($D_n$ is kept fixed). Motivated by the goal of preserving validity and improving efficiency, we now expand on a method to construct $\Gamma^{(\alpha)}$ known as weighted majority vote.

### 3.1 WEIGHTED MAJORITY VOTE (WMV)

This approach was originally proposed by (Cherubin, 2019) and it constructs $\Gamma^{(\alpha)}$ by including every $y \in \mathcal{Y}$ that is present in the majority of the prediction sets (i.e. $y \in \Gamma^{(\alpha)} \iff \sum_{m=1}^{M} \frac{1}{M} \mathbf{1}\{y \in C_m^{(\alpha)}(X)\} > 1/2$). We refer to this initial construction as majority vote (**MV**). To generalize MV,

---

[1]The non-conformity score we use is $1 - f_m(X_i)_{Y_i}$ where $f_m(X_i)_{Y_i}$ corresponds to the model estimate of $P(Y_i|X_i)$. For instance, if $f_m$ is a neural network, the score is one minus the softmax output of the correct class (Angelopoulos & Bates, 2022)

we can parametrize the weight each conformal predictor gets, leading to weighted majority vote:

$$\Gamma_w^{(\alpha)}(X) = \{y \in Y : \sum_{m=1}^{M} w_m \mathbf{1}\{y \in C_m^{(\alpha)}(X)\} > 1/2, w \in \Delta\} \tag{1}$$

$$\Delta = \{w \in R_+^M : \sum_{m=1}^{M} w_m = 1\} \tag{2}$$

As the name suggests, a label $y \in \mathcal{Y}$ is in $\Gamma_w^{(\alpha)}$ if it is present in the "weighted majority" of the conformal predictors (i.e. $y \in \Gamma^{(\alpha)} \iff \sum_{m=1}^{M} w_m \mathbf{1}\{y \in C_m^{(\alpha)}(X)\} > 1/2$)). Following results from Gasparin & Ramdas (2024b), equation (1) guarantees $\mathbb{P}(Y_{n+1} \notin \Gamma_w^{(\alpha)}(X_{n+1}) \leq 2\alpha$ for all $w \in \Delta$ and thus validity is preserved if we reconstruct the collection of prediction sets at a more conservative error level (i.e. $C_m^{(\alpha/2)}$ instead of $C_m^{(\alpha)}$). Nonetheless, the issue with more conservative sets is that they tend to be larger (i.e. $|C_m^{(\alpha/2)}(X)| \geq |C_m^{(\alpha)}(X)|$ for all $m \in [M]$) and thus inappropriate choices of $w$ can render aggregation useless (i.e. there exists $m \in [M]$ such that $\mathbb{E}_X|\Gamma_w^{(\alpha/2)}(X)| \geq \mathbb{E}_X|C_m^{(\alpha)}(X)|$). For instance, in Figure 1 naively choosing MV (i.e. $w = (1/M, ..., 1/M)$) results in larger prediction sets than choosing the estimated by EWMV. Accordingly, in the next section we propose an approach to estimate the aggregation weights $w$ in a data driven way so as to mitigate the efficiency issue.

## 4 Estimating efficient weights for WMV

Given the WMV aggregation algorithm, the optimal aggregation weights are:

$$w^* = \arg\min_{w \in \Delta} \mathbb{E}_X|\Gamma_w^{(\alpha/2)}(X)| \tag{3}$$

To approximate $\mathbb{E}_X|\Gamma_w^{(\alpha/2)}(X)|$ in equation equation 3, we employ a sample $D_{n_{\text{est}}} := (X_i)_{i=1}^{n_{\text{est}}} \overset{\text{iid}}{\sim} \mathbb{P}_X$, separate from the calibration sample $D_n$, and perform empirical risk minimization (ERM):

$$\hat{w} = \arg\min_{w \in \Delta} \frac{1}{n_{\text{est}}} \sum_{i=1}^{n_{\text{est}}} |\Gamma_w^{(\alpha/2)}(X_i)| \tag{4}$$

Assuming $\mathcal{Y}$ countable, we can compute cardinality with the counting measure $|\Gamma_w^{(\alpha/2)}(X)| = \sum_{y \in \mathcal{Y}} \mathbf{1}\{y \in \Gamma_w^{(\alpha/2)}(X)\}$. By plugging this into equation equation 4 and replacing $\Gamma_w^{(a/2)}(X)$ with equation equation 1, our optimization problem becomes:

$$\hat{w} = \arg\min_{w \in \Delta} \frac{1}{n_{\text{est}}} \sum_{i=1}^{n_{\text{est}}} \sum_{y \in \mathcal{Y}} l_i^{(y)}(w) \quad \text{s.t. } l_i^{(y)}(w) = \mathbf{1}\left\{\sum_{m=1}^{M} w_m \mathbf{1}\{y \in C_m^{(\alpha/2)}(X_i)\} > \frac{1}{2}\right\} \tag{5}$$

Now we delve into two strategies to solve the optimization problem (5).

### 4.1 Mixed integer linear program formulation (MILP)

We reformulate optimization problem (5) as an MILP and let $\delta_i^{(y)} = \mathbf{1}\{\sum_{m=1}^{M} w_m \mathbf{1}\{y \in C_m^{(\alpha/2)}(X_i)\} > 1/2\}$ play the role of $l_i^{(y)}(w)$. We refer to this as **MILP**

$$\hat{w}_{\text{MILP}} = \arg\min_{\substack{w \in \Delta \\ \delta_i^{(y)} \in \{0,1\}}} \sum_{i=1}^{n_{\text{est}}} \sum_{y \in \mathcal{Y}} \delta_i^{(y)} \quad \text{s.t. } \delta_i^{(y)} \geq \sum_{m=1}^{M} w_m \mathbf{1}\{y \in C_m^{(\alpha/2)}(x_i)\} - \frac{1}{2} \tag{6}$$

### 4.2 Linear program formulation (LP)

Unfortunately, the MILP reformulation equation 6, in the worst case, can result in exhaustive search. Accordingly, we relax it into a convex problem by approximating the outmost indicator function with a hinge loss; we then reformulate it as a linear program using the epigraph trick and refer to the solution as **LP**.

$$\hat{w}_{\text{LP}} = \arg\min_{\substack{w \in \Delta \\ t \geq 0}} \sum_{i=1}^{n_{\text{est}}} \sum_{y \in \mathcal{Y}} t_i^{(y)} \quad \text{s.t. } t_i^{(y)} \geq \sum_{m=1}^{M} w_m \mathbf{1}\{y \in C_m^{(\alpha/2)}(x_i)\} - \frac{1}{2} \tag{7}$$

## 5 AGGREGATION ALGORITHM: EWMV

In practice, we generally do not have direct access to the collection of conformal predictors $(C_m^{(\alpha/2)})_{m \in [M]}$. Instead, we have access to a calibration dataset $D_n$, a collection of pre-estimated classifiers $(f_m : \mathcal{X} \to \mathcal{Y})_{m \in [M]}$, a user-specified error level $\alpha$ and a conformal method (CM). We assume the conformal method (CM) constructs a valid conformal predictor $C_m^{(\alpha)}$ using the corresponding $f_m$ classifier to produce the non-conformity scores for $D_{n+1}$.

We propose Algorithm 1 (EWMV) to estimate aggregation weights $\hat{w}$ and produce the aggregated conformal predictor $\Gamma_{\hat{w}}^{(\alpha/2)} : \mathcal{X} \to 2^{\mathcal{Y}}$ with desired error level $\alpha$.

---

**Algorithm 1** (EWMV)

**Input:** i.i.d. sample $(D_n)$, collection of classifiers $(f_m : \mathcal{X} \to \mathcal{Y})_{m \in [M]}$, conformal method (CM), error level $\alpha \in (0, 1)$

$D_{n_{\text{est}}}, D_{n_{\text{cal}}} \leftarrow \text{Split}(D_n)$

**for** $m = 1$ **to** $M$ **do**

    **for** $x_i \in D_{n_{\text{est}}}$ **do**

        $C_m^{(\alpha/2)}(x_i) \leftarrow \text{CM}(f_m(\cdot), x_i, D_{n_{\text{cal}}}, \alpha/2)$

    **end for**

**end for**

$\hat{w} \leftarrow \{\text{LP or MILP}\}(C_m^{(\alpha/2)}(x_i) \,\forall\, x_i \in D_{n_{\text{est}}}, m \in [M])$

**return** $\hat{w}$

---

**Proposition 5.1.** *Let $D_{n+1}$ be an i.i.d. sample, let $\mathbb{P}(Y_{n+1} \notin C_m^{(\alpha/2)}(X_{n+1})) \leq \alpha/2$ for every $m \in [M]$, and let $\hat{w}$ be estimated by Algorithm 1 on a hold-out set. It then follows that for a set $\Gamma_{\hat{w}}^{(\alpha/2)}(X_{n+1})$ constructed using equation equation 1:*

$$\mathbb{P}(Y_{n+1} \notin \Gamma_{\hat{w}}^{(\alpha/2)}(X_{n+1})) \leq \alpha. \tag{8}$$

*Proof in appendix section A.1.* □

We emphasize that EWMV, in essence, returns a set-valued function (i.e. $\Gamma_{\hat{w}}^{(\alpha/2)}$) and not the specific prediction set of a given input (i.e. $\Gamma_{\hat{w}}^{(\alpha/2)}(X)$). In the case EWMV returns an indicator vector (i.e. $\hat{w} = e^{(i)}$), the indicated conformal predictor at level $(\alpha)$ should be used (i.e. $C_i^{(\alpha)}$) instead of $\Gamma_{\hat{w}}^{(\alpha/2)}$. Lastly, if the average size of the most efficient predictor does not change when we re-estimate it at a more conservative level, in the limit of estimation samples, we expect EWMV will provide a valid predictor that is as efficient or better. Proposition A.1 establishes this. This can materialize when the distribution of the non-conformity score is discrete (e.g. in risk stratification García et al. (2024)).

## 6 EXPERIMENTS

In these experiments we measure the efficiency and validity of EWMV (Algorithm 1) on four tasks: multi-class classification on synthetic data, image classification, risk stratification and natural question answering. For each task we collect a dataset, a conformal method and a multitude of pre-estimated predictors. We then split the data randomly into a calibration set $(D_{n_{\text{cal}}})$, an estimation set $(D_{n_{\text{est}}})$ and a test set. Given a validity limit $\alpha$, we perform EWMV to estimate the aggregation weights $\hat{w}$ using both the MILP equation 6 and LP equation 7 formulations (See appendix figure 15). Lastly, we measure the empirical validity (i.e. $\frac{1}{n_{\text{test}}} \sum_{i=1}^{n_{\text{test}}} \mathbf{1}\{y_i \in \Gamma_{\hat{w}}^{(\alpha/2)}(x_i)\}$) and empirical efficiency (i.e. $\frac{1}{n_{\text{test}}} \sum_{i=1}^{n_{\text{test}}} |\Gamma_{\hat{w}}^{(\alpha/2)}(x_i)|$) of the corresponding combined set. We compare against every recomputed prediction sets $(C_m^{(\alpha)})_{m \in [M]}$ at error level $\alpha$ using the entire available dataset (i.e. $D_n = D_{\text{est}} \cup D_{\text{cal}}$). The reason for the recomputation is to provide a fair comparison with respect to not doing aggregation and instead performing standard conformal prediction over an individual

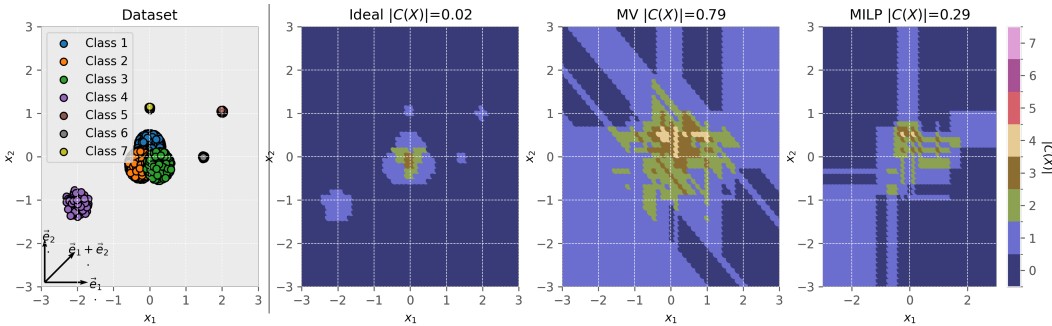

Figure 2: (left-most) Synthetic 2D dataset with color coded classes. (middle-left) Ideal prediction set size. (middle-right) Estimated prediction set size with MV. (right-most) Estimated prediction set size using MILP.

predictor (See appendix figure 14). Note that for the synthetic (section 6.1) and risk stratification (section 6.4) experiments, we further split the estimation data ($D_{\text{est}}$) into a training set ($D_{\text{train}}$) for model training.

## 6.1 SYNTHETIC EXPERIMENT

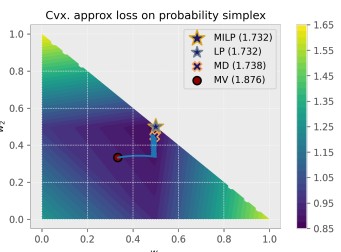

Figure 3: Loss landscape formulation equation 7 w.r.t. the probability simplex. Marked are the estimated weights.

The goal is to qualitatively assess the performance of the proposed weight estimation methods on a multimodal setting. Consider the 4K sample dataset in the leftmost plot of Figure 2; we split it into 500 samples for calibration ($D_{\text{cal}}$), 500 samples for estimation ($D_{\text{est}}$), 1K sample for training ($D_{\text{train}}$), and 2K samples for testing. We synthesize the multimodal setting by projecting each 2D-input onto three linear subspaces: $\vec{e_1}, \vec{e_2}, \vec{e_1} + \vec{e_2}$. Three MLP classifiers are then trained with each separate subspace projection. The ideal plot color codes the size of the oracle prediction set for every input in the 2D-input space. The goal of an aggregation algorithm is to get qualitatively "close" to the ideal plot without compromising validity beyond $\alpha = 0.05$. For each method, we estimate valid conformal sets (i.e. $C_1(X), C_2(X), C_3(X)$) with the adaptive prediction sets (APS) algorithm (Angelopoulos & Bates, 2022). We then aggregate the $C_{1:3}$ using Algorithm 1 with **MILP** optimization. In Figure 2 (middle-right) we color code the size of the prediction sets produced by **MV** (see section 3.1). In Figure 2 (right-most) we color code the size of the sets from the **MILP** method. We qualitatively observe the **MILP** set (0.29) is on average smaller than the **MV** set (0.79) and that both are valid. In turn, **MILP** is closer to the ideal performance. This supports the hypothesis that data-driven parametrization of the WMV algorithm can result in efficiency gains. Interestingly, we observe that the discrepancy between prediction sets result in empty sets. We speculate this discrepancy between sets is connected to discrepancy between predictors and, accordingly, could inform the epistemic uncertainty of a point (Hüllermeier & Waegeman, 2021). This follows from not having data around that point to ground different predictors to a specific label. Lastly, Figure 3 also explains the relationship between **MV** and **LP**. In particular, we can solve the constraint convex optimization problem equation 7 with Mirror Descent (Nemirovskij & Yudin, 1983). The optimization weights are initialized at **MV** and, given an appropriate step-size, iteratively converge to **LP**. Accordingly, w.r.t. the convex loss and the estimation set, **LP** is a better solution than **MV**. Surprisingly, **LP** and **MILP** coincide.

## 6.2 APPLICATION: IMAGE CLASSIFICATION

The goal is to correctly classify images from CIFAR-100 (Krizhevsky, 2009) and Imagenet (Russakovsky et al., 2015) datasets. We use RAPS from Angelopoulos et al. (2022) as the conformal

| Name | Inefficiency (↓) | Validity (≥ .90) |
|------|------|------|
| AlexNet+RAPS | $13.81 \pm 0.97$ | $0.899 \pm 0.005$ |
| SqueezeNet+RAPS | $11.67 \pm 0.45$ | $0.900 \pm 0.005$ |
| MobileNet+RAPS | $8.24 \pm 0.27$ | $0.900 \pm 0.006$ |
| Resnet50+RAPS | $6.98 \pm 0.28$ | $0.900 \pm 0.005$ |
| Inception+RAPS | $6.32 \pm 0.11$ | $0.900 \pm 0.005$ |
| VGG+RAPS | $4.10 \pm 0.11$ | $0.900 \pm 0.007$ |
| ConvnNext+RAPS | $3.67 \pm 0.24$ | $0.901 \pm 0.008$ |
| Resnet+RAPS | $3.30 \pm 0.10$ | $0.900 \pm 0.008$ |
| DenseNet+RAPS | $3.18 \pm 0.10$ | $0.901 \pm 0.007$ |
| Swin+RAPS | $2.46 \pm 0.07$ | $0.900 \pm 0.005$ |
| Regnet+RAPS | $2.41 \pm 0.06$ | $0.900 \pm 0.006$ |
| DinoV2+RAPS | $2.14 \pm 0.03$ | $0.901 \pm 0.006$ |
| ViT+RAPS | $1.76 \pm 0.04$ | $0.900 \pm 0.005$ |
| MV | $3.46 \pm 0.16$ | $0.975 \pm 0.002$ |
| LP (Ours) | $1.86 \pm 0.28$ | $0.964 \pm 0.009$ |
| **MILP (Ours)** | $1.54 \pm 0.14$ | $0.916 \pm 0.008$ |

Table 1: Inefficiency and validity of multiple conformal predictors on Imagenet. Experiment is repeated 10 times on random splits of the data and we report $\mu \pm 2\sigma$.

| Name | Inefficiency (↓) | Validity (≥ .90) |
|------|------|------|
| Resnet50+RAPS | $2.98 \pm 0.19$ | $0.899 \pm 0.015$ |
| Swin-tiny-p4w7+RAPS | $2.84 \pm 0.15$ | $0.899 \pm 0.014$ |
| ConvNext+RAPS | $2.19 \pm 0.13$ | $0.898 \pm 0.016$ |
| Swin-tiny+RAPS | $2.06 \pm 0.06$ | $0.902 \pm 0.008$ |
| Swin-small+RAPS | $1.66 \pm 0.06$ | $0.898 \pm 0.011$ |
| ViT-base+RAPS | $1.51 \pm 0.06$ | $0.898 \pm 0.009$ |
| ViT-large+RAPS | $1.38 \pm 0.07$ | $0.899 \pm 0.015$ |
| ViT+RAPS | $1.32 \pm 0.07$ | $0.900 \pm 0.018$ |
| Swin-base+RAPS | $1.29 \pm 0.03$ | $0.901 \pm 0.014$ |
| ViT-base-in21k+RAPS | $1.28 \pm 0.05$ | $0.900 \pm 0.012$ |
| MV | $1.44 \pm 0.04$ | $0.982 \pm 0.003$ |
| LP (Ours) | $1.20 \pm 0.14$ | $0.950 \pm 0.058$ |
| **MILP (Ours)** | $1.14 \pm 0.13$ | $0.910 \pm 0.011$ |

Table 2: Inefficiency and validity of multiple conformal predictors on CIFAR-100. Experiment is repeated 10 times on random splits of the data and we report $\mu \pm 2\sigma$.

method and obtain all the fine-tuned models along with the dataset are available from HuggingFace and Torchvision. We split the dataset into two-thirds for testing $D_{\text{test}}$ and one-third for calibration $D_{\text{n}}$. To evaluate methods (i.e. MV, LP and MILP), we further split the calibration dataset $D_n$ into 90% for calibration $D_{\text{cal}}$ and 10% for estimation $D_{\text{est}}$. CIFAR-100 results are on Table 2 and Imagenet results are on Table 1. Both suggest MILP is more efficient than any individual, or aggregated, conformal predictor. Furthermore, its validity is closer to nominal levels than MV or LP. The reason why aggregation methods have larger validity is because the individual predictors are estimated at a more conservative error-level (i.e. $\alpha/2$). It is also interesting to note that adding models tends to benefit aggregation efficiency. We expect this is because the estimated weights tease out the most efficient models to aggregate. This last point is further explored in Section 6.8.

### 6.3 APPLICATION: NATURAL QUESTION ANSWERING

For this experiment we closely follow the setup from Li et al. (2024). The goal is to correctly answer a query using a collection of passages from Wikipedia. We use the TRAQ (Li et al., 2024) as the conformal method. In short, this method applies standard conformal prediction in two stages: (1) to construct a prediction set of passages from a retriever model; (2) to construct a set of answers associated with each passage from an LLM. The final prediction set corresponds to the union of the answers sets of all passages. The main difficulty arises in determining when the true answer $y$ is in the set $C_m^{(\alpha)}(X)$, due to the multitude of semantically similar words that could arise. Accordingly, like Li et al. (2024), we consider $y \in C_m^{(\alpha)}(X)$ when $\exists_{e \in C_m^{(\alpha)}(X)}(\text{rouge-1}(y, e) > 0.3)$ and where the rouge-1 score measures semantic similarity (Lin, 2004). In terms of the architecture, we utilize the Dense Passage Retriever (DPR) from Karpukhin et al. (2020) as a retriever model and a variety of LLMs from Huggingface as predictors. We evaluate these methods using 560 queries from the Natural questions dataset (Kwiatkowski et al., 2019) and use the WikiDPR dataset for passages (Karpukhin et al., 2020). We randomly split the data into calibration (35%), estimation (35%) and testing (30%). We compute the TRAQ prediction sets for multiple models with both the calibration and estimation splits setting $\alpha = 0.2$ as the validity limit. We then use EWMV to compute the aggregation weights with the estimation split. We repeat this experiment 10 times and report the validity and efficiency in Table 3. We observe that the MV method yields combination useless. Nonetheless, both the LP and MILP method improve efficiency without compromising validity. It is important to note that, like Li et al. (2024), the prediction sets can be quite large (approx. 30 answers) to guarantee validity. Li et al. (2024) recommends semantic clustering to remove redundancies during deployment.

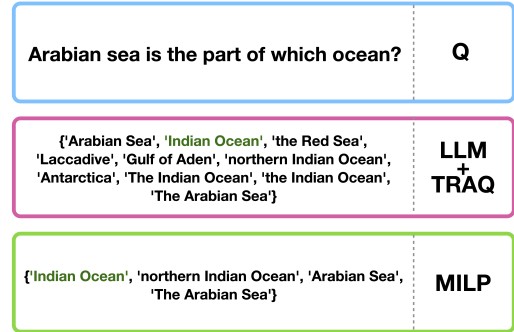

Figure 4: Smallest LLM prediction set (**LLM+TRAQ**) and from proposal (**MILP**) for a given query **Q**. Green indicates correct answer.

| Model | Inefficiency ($\downarrow$) | Validity ($\geq 0.80$) |
|---|---|---|
| MiniLM+TRAQ | $29.53 \pm 5.0$ | $0.84 \pm 0.1$ |
| DynamicBert+TRAQ | $28.72 \pm 5.0$ | $0.85 \pm 0.1$ |
| Roberta+TRAQ | $28.68 \pm 5.0$ | $0.87 \pm 0.1$ |
| DistillBert+TRAQ | $27.33 \pm 4.5$ | $0.86 \pm 0.1$ |
| MobileBert+TRAQ | $27.25 \pm 4.4$ | $0.86 \pm 0.1$ |
| MV | $29.34 \pm 3.3$ | $0.90 \pm 0.1$ |
| LP (Ours) | $22.22 \pm 2.3$ | $0.89 \pm 0.1$ |
| **MILP** (Ours) | $18.01 \pm 1.9$ | $0.86 \pm 0.1$ |

Table 3: LLM experiment. Experiment is repeated 10 times on random splits of the data. We report $\mu \pm 2\sigma$.

## 6.4 APPLICATION: ACUTE CORONARY SYNDROME (ACS) RISK STRATIFICATION

The goal is to correctly stratify ACS cases as high/low risk while minimizing the number of intermediate risk cases (García et al., 2024). The dataset contains 3300 samples for training, calibration, estimation, and 400 samples for testing. The models to be aggregated are GBDT (Malinin et al., 2021), FR (Liu et al., 2022), and ECG-DL (Xiao et al., 2022). The setting is multi-modal, as each case has a collection of signs and symptoms processed by GBDT and FR, and a ECG trace processed by ECG-DL model. Prediction sets are estimated using class-conditional conformal estimation (CC) (Lei, 2014). Risk stratification performance is measured in terms of definitive percentage (i.e. proportion of prediction sets that are either $\{0\}$ or $\{1\}$) and balanced accuracy (BACC) performance (i.e. (sensitivity + specificity)/2). The higher the definitive percentage and the BACC, the better. The validity limit is set to 5% (i.e. $\alpha = .05$). The results in Table 4 suggest that LP is the most efficient of the aggregation methods and reasonably exceeds the validity limit per chapter three in Angelopoulos & Bates (2023). In the context of early ACS detection, as long as validity stays within set limits, greater efficiency increases definitive percentages, and thus reduces resource misallocation and prevents delays in time-sensitive therapies.

## 6.5 WHAT IS THE RUNTIME COMPLEXITY OF EWMV?

The worst case runtime complexity of EWMV depends on the specific optimization formulation. Consider $n_{est}$ to be the number of estimation samples in $D_{n_{est}}$ and $|\mathcal{Y}|$ to be the cardinality of our label space. In the worst case, the time complexity of **MILP** is exponential in this product (i.e. $\mathcal{O}(e^{|\mathcal{Y}| \times n_{est}})$); and for **LP**, the worst time complexity is polynomial (i.e. $\mathcal{O}(W(|\mathcal{Y}| \times n_{est})^{1/2} + (|\mathcal{Y}| \times n_{est})^{5/2})$), where $nnz(A) < W$, $A \in \{0,1\}^{n_{est} \times |\mathcal{Y}|}$ and $A_{ij} = \mathbf{1}\{y_j \in C(x_i)\}$ per Lee & Sidford (2015). In figure 5, we empirically assess the runtime of MILP in seconds (s) across a variety of $|\mathcal{Y}| \times n_{est}$ products. When $|Y| \times n_{est} = 200K$, on V2-8 TPU, the runtime is 12m and 7m for MILP and LP respectively, with the runtime rate of MILP growing faster than LP as expected. For reference, we also plot the runtime of MD (From section 6.1) with a fixed number of iterations. As opposed to LP and MILP, MD requires hyperparameter tuning to work.

| Method | Validity ($\geq 95$) | Inefficiency ($\downarrow$) | Definitive-% ($\uparrow$) | BACC ($\uparrow$) |
|---|---|---|---|---|
| FR+CC | $100 \pm 0$ | $1.62 \pm 0.22$ | $38 \pm 22$ | $100 \pm 1$ |
| GBDT+CC | $99 \pm 3$ | $1.46 \pm 0.25$ | $54 \pm 25$ | $94 \pm 19$ |
| ECG-DL+CC | $98 \pm 1$ | $1.94 \pm 0.07$ | $6 \pm 7$ | $64 \pm 40$ |
| MV | $99 \pm 1$ | $1.64 \pm 0.20$ | $36 \pm 20$ | $99 \pm 2$ |
| LP (Ours) | $94 \pm 1$ | $1.32 \pm 0.10$ | $65 \pm 10$ | $96 \pm 4$ |
| **MILP** (Ours) | $94 \pm 1$ | $1.32 \pm 0.10$ | $65 \pm 10$ | $96 \pm 4$ |

Table 4: Risk stratification experiment. Experiment is repeated 10 times on random splits of the data. We report $\mu \pm 2\sigma$

## 6.6 COMPARING EWMV WITH CONFORMAL AGGREGATION BASELINES

We compare EWMV with multiple p-value methods from section 2 (i.e. $\frac{K}{k} p_{(k)}$ (Rüger, 1978), Average ($2\bar{p}$) (Rüschendorf, 1982)) on the task of image classification on CIFAR-100 with $\alpha = 0.05$. In table 5, $k$ parametrizes the corrected k'th ordered p-value $\frac{K}{k} p_{(k)}$ approach from Rüger (1978) where $K$ represents the number of models. Per Gasparin & Ramdas (2024a), $k = 1$ recovers Bon-

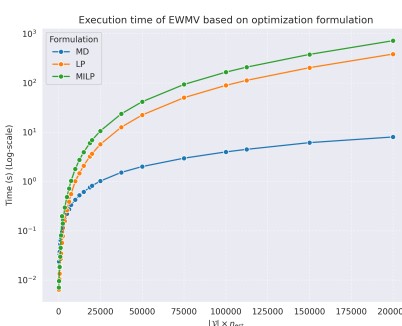

Figure 5: Runtime in seconds (s) of EWMV for different formulations across multiple label space sizes and estimation dataset sizes.

| Name | Inefficiency ($\downarrow$) | Validity ($\geq .95$) |
|---|---|---|
| $k = 10$ (Union) | $7.95 \pm 0.32$ | $0.999 \pm 0.001$ |
| Average | $4.84 \pm 0.24$ | $0.998 \pm 0.001$ |
| YKadj | $2.97 \pm 0.34$ | $0.983 \pm 0.006$ |
| YKsplit | $2.68 \pm 1.74$ | $0.949 \pm 0.012$ |
| YK | $2.00 \pm 0.12$ | $0.951 \pm 0.010$ |
| MV | $1.84 \pm 0.08$ | $0.991 \pm 0.002$ |
| $k = 5$ | $1.84 \pm 0.09$ | $0.991 \pm 0.002$ |
| Bonferroni | $1.81 \pm 0.31$ | $0.974 \pm 0.008$ |
| Heuristic | $1.76 \pm 0.08$ | $0.992 \pm 0.002$ |
| $k = 3$ | $1.63 \pm 0.13$ | $0.984 \pm 0.003$ |
| Single+RAPS | $1.58 \pm 0.07$ | $0.950 \pm 0.007$ |
| LP (Ours) | $1.44 \pm 0.22$ | $0.979 \pm 0.020$ |
| **MILP (Ours)** | $1.32 \pm 0.08$ | $0.957 \pm 0.007$ |
| Fisher* | $0.99 \pm 0.01$ | $0.914 \pm 0.005$ |

Table 5: Efficiency and validity of various conformal combination approaches from section 2. *Fisher does not meet validity.

ferroni correction, $k = 5$ recovers MV, $k = 10$ recovers set union and $k = 3$ is the most efficient of the $k$ values. For reference, we also include the most efficient individual model conformalized at level ($\alpha = .05$) with all the calibration data (i.e. Single+RAPS), WMV with a heuristic weight (e.g. weights inversely proportional to the empirical size), and the Fisher p-value method (Balasubramanian et al., 2015) included for completeness. We also consider three methods from Yang & Kuchibhotla (2025) (i.e. YK, YKadj, YKsplit) which correspond to the standard, adjusted and data-split proposal to perform model selection Liang et al. (2025). Results in table 5 showcase EWMV as the only valid aggregation method more efficient than the best individual model (i.e. Single+RAPS). Fisher's method cannot guarantee validity because it assumes independence among the p-values being aggregated and all these p-values depend on the same random variable $X_{n+1}$.

### 6.7 IS THERE A THEORETICAL UPPER BOUND ON THE COVERAGE OF EWMV?

Coverage is the probability the correct answer is in (i.e. $\mathbb{P}(Y_{n+1} \in \Gamma_{\hat{w}}^{(\alpha/2)}(X_{n+1}))$). There is not a practical theoretical upper bound for the coverage of EWMV. Gasparin & Ramdas (2024b) prove an upper bound for the coverage of WMV when the weights are uniform (Theorem 2.5 from Gasparin & Ramdas (2024b)). Unfortunately the bound becomes meaningless when aggregating more than two models at commonplace error-levels $\alpha < 0.25$. Generalizing this bound beyond uniform weights is not trivial because it involves the analysis of a weighted sum of dependent indicator random variables variables (See definition of $\Gamma_w^{(\alpha)}$ in equation 1). That said, some conformal methods (e.g. RAPS and APS) approach nominal

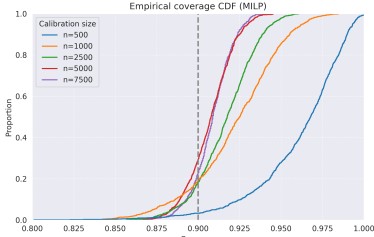

Figure 6: Empirical coverage estimate of EWMV across multiple calibration sample sizes $n_{\text{cal}}$ for ($\alpha = 0.1$) using RAPS as the conformal method aggregating three CIFAR-10 predictors.

coverage as we increase the number of calibration samples and it begs the question: Does EWMV approach nominal coverage? In figure 6 this appears to be the case albeit at a slower rate than the theoretical rate of the individual predictor (appendix figure 10). Curiously, we observe this is not the case for uniform weights (i.e. MV) in appendix figure 11. Please refer to appendix section A.3.5 for more details.

### 6.8 HOW DOES THE NUMBER OF MODELS AFFECT THE INEFFICIENCY OF EWMV AND OTHER BASELINES?

For this experiment, we repeat the CIFAR-100 classification experiment from section 6.2 ten times, each with a corresponding number of models to combine. The goal is to measure the impact the

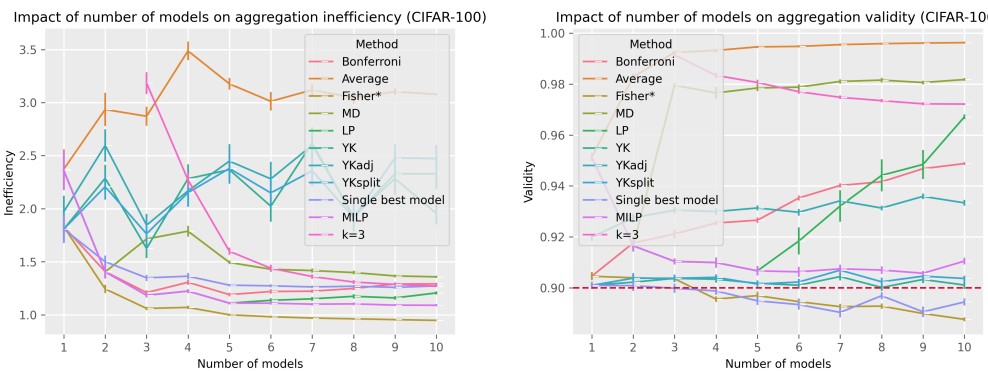

Figure 7: Inefficiency (left) and validity (right) of the most efficient collection of aggregation baselines from 6.6 as a function of the number of models, reported is $\mu \pm 2\sigma$ over ten runs, with error-level $\alpha = 0.1$. Of the approaches that preserve validity, MILP and LP are the most efficient.

number of models play on the efficiency of EWMV and the baselines from section 6.6. For each model size, we randomly sample the corresponding number of models from the ten models available. For reference, we also report the performance of the single best model in the sample. Results under Figure 7 indicate efficiency improves, on average, the more models we add. We further observe these benefits diminish as we consider more models. This results suggest EWMV estimates the more efficient combination the more models we consider and thus makes an argument for collecting more models. The diminishing returns also suggest a sparse combination may provide a reasonable efficiency/compute tradeoff. This is particularly important for tasks where model inference is costly (e.g. Open QA).

# 7 CONCLUSIONS & FUTURE WORK

In this work we propose EWMV, a novel algorithm to improve the efficiency of conformal methods by leveraging two readily available resources: the calibration data and a plethora of pre-estimated predictors. We show EWMV leads to more efficient conformal predictors for image classification, natural question answering and risk stratification. This is important because reducing the size of the prediction sets, without compromising validity, mitigates false discovery costs in drug discovery and delayed response of medical emergencies. Future work could explore aggregation of conditionally valid conformal predictors to ensure coverage of relevant groups; furthermore, it may open up the possibility to tailor the weights according to the input, rather than having uniform weights across the space. It could also be fruitful to explore weight estimation to aggregate risk controlling prediction sets, as this has the potential to mitigate inefficiencies in other tasks (e.g. image segmentation).

**Reproducibility statement**: All details to reproduce experiments are in appendix section A.3

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

Hao Zeng, Kangdao Liu, Bingyi Jing, and Hongxin Wei. Parametric Scaling Law of Tuning Bias in Conformal Prediction.

# A    APPENDIX

## A.1    PROOFS

*Proof of Proposition 5.1.* The solution space for the aggregation weights $w$ to Problems equation 6 and equation 7 is the probability simplex. Accordingly, the estimated weight will satisfy $\hat{w} \in \Delta$. Following the same strategy for Theorem 2.1 from Gasparin & Ramdas (2024b), validity is guaranteed by Markov's inequality, as well as the linearity and monotonicity of expectation. $\qquad\square$

**Proposition A.1.** *Let $\alpha \in [0, 1]$, $t \in [0, 1]$ represent the aggregation threshold (e.g. $t = 1/2$) and $\underline{m} = \arg\min_{m \in [M]} \{\mathbb{E}_X |C_m^{(\alpha)}(X)|\}$ where $|\cdot|$ represents set cardinality. If $\mathbb{E}_X |C_{\underline{m}}^{(\alpha(1-t))}(X)| = \mathbb{E}_X |C_{\underline{m}}^{(\alpha)}(X)|$ then $\mathbb{E}_X |\Gamma_{w^*}^{(\alpha(1-t))}(X)| \leq \mathbb{E}_X |C_m^{(\alpha)}(X)|$ for all $m \in [M]$ where $\Gamma_{w^*}^{(\alpha(1-t))}$ is constructed with equation 1 .*

*Proof in appendix section A.1.* $\qquad\square$

*Proof of Proposition A.1.* Let $\underline{w} \in \Delta$ correspond to a weight vector with unit mass at index $\underline{m}$ then $\mathbb{E}_X |\Gamma_{w^*}^{(\alpha(1-t))}(X)| \leq \mathbb{E}_X |\Gamma_{\underline{w}}^{(\alpha(1-t))}(X)| = \mathbb{E}_X |C_{\underline{m}}^{(\alpha(1-t))}(X)| = \mathbb{E}_X |C_{\underline{m}}^{(\alpha)}(X)| \leq \mathbb{E}_X |C_m^{(\alpha)}(X)|$ for all $m \in [M]$. The first inequality follows from formulation equation 3, the next two equalities follows from equation equation 1 and the assumption above respectively, and the last inequality follows from the definition of $\underline{m}$. $\qquad\square$

## A.2    EXTRA EXPERIMENTS

### A.2.1    WHAT IS THE RECOMMENDED NUMBER OF CALIBRATION/ESTIMATION SAMPLES?

Our approach relies on sufficient data for an estimation split. How much will be problem dependent but we observe that with ($n_{\text{est}} < 500$) estimation samples MILP/LP is able to estimate (in $<$30m on a v2-8 TPU) a more efficient predictor on CIFAR-100, Imagenet and Natural QA experiments. We recommend to collect the remaining number of calibration samples ($n_{\text{cal}}$) per the guidelines from section 3.2 Angelopoulos & Bates (2022). Please see figure 15 for references to $n_{\text{est}}$ and $n_{\text{cal}}$.

### A.2.2    MITIGATING THE NEED FOR ESTIMATIONS SAMPLES

The work of Zeng et al. explores the coverage bias induced by using the calibration data for estimation of parametric approaches for conformal prediction (e.g. Angelopoulos et al. (2022); Luo & Zhou). We repeat the CIFAR-10 experiment done by Zeng et al., and use the entire calibration dataset for both estimation and calibration (same) and compare it to using the estimation split for estimation (hold-out). Interestingly, we did not observe a statistically significant difference in the coverage gap between "same" and "hold-out". This provides some preliminary evidence to mitigate the need of "hold-out" data. Future work could extend the analysis of Zeng et al. to determine if the coverage gap can be asymptotically bounded.

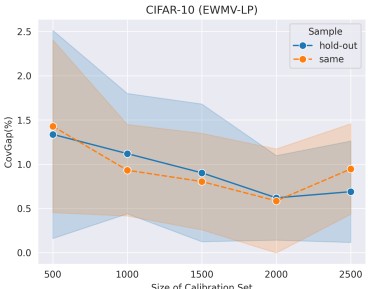

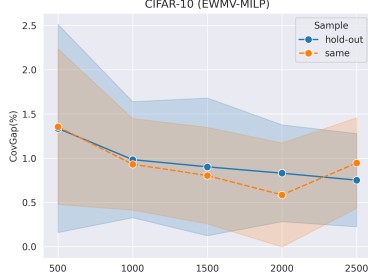

Figure 8: Coverage gap for LP formulation of EWMV.

Figure 9: Coverage gap for MILP formulation of EWMV.

### A.3 EXPERIMENT DETAILS

#### A.3.1 SYNTHETIC EXPERIMENT: SECTION 6.1

This experiment is run on a Intel(R) Xeon(R) CPU E5-2683 v4 with 32Gb of memory and 125Gb of Disk. The Synthetic dataset is attached. Data is randomly split into calibration, estimation, training and test sets. We train three MLPs for using 200 batch size for 300 epochs with 0.01 learning rate. Each MLP has 500 hidden units. Once all models are trained, we apply APS (Angelopoulos & Bates, 2022) and our proposed Algorithm 1. We use Gurobi to solve the proposed LP and MILP, that is formulations equation 7 and equation 6 respectively). Total runtime is ($< 2$ hours). Both LP and MILP estimation is ($< 10$ minutes).

#### A.3.2 IMAGE CLASSIFICATION EXPERIMENTS: SECTION 6.2

The Imagenet validation dataset can be obtained from (Wolf et al., 2020). All the pre-estimated models listed in Table 1 can be downloaded from (Wolf et al., 2020). The RAPS conformal implementation is taken from (Angelopoulos et al., 2022). For each pre-estimated model, we perform inference over the entire dataset using on an A-100 GPU and save the softmax outputs. To apply our proposal, we load the softmax scores, randomly split the scores into test, calibration and estimation sets, and perform Algorithm 1 on a v2-8 TPU with 300Gb of memory and 225Gb of Disk. We use Gurobi to solve the proposed LP and MILP, that is formulations equation 7 and equation 6 respectively. We repeat this experiment ten times, each with a different random split. Total runtime is ($< 10$ hours). Both LP and MILP estimation is ($< 30$ minutes) for each repetition. This process is the same for the CIFAR-100 dataset (Available to download from Wolf et al. (2020)). With corresponding models under table 2 available for download. Please refer to table under A.5 for corresponding URLS.

#### A.3.3 OPEN DOMAIN QA EXPERIMENT: SECTION 6.3

We follow the instructions in https://github.com/shuoli90/TRAQ to download the dataset (Kwiatkowski et al., 2019) and apply the TRAQ conformal method (Li et al., 2024) on the listed language models from Table 3. All the LMs are publicly available to download from (Wolf et al., 2020). We perform model inference on the entire dataset, obtain the TRAQ prediction sets and apply Algorithm 1 on a M2 Mac Studio with 32Gb of memory and 500Gb of disk. We repeat this experiment ten times, each with a different random split of scores into test, calibration and estimation sets. We use Gurobi to solve the proposed LP and MILP, that is formulations equation 7 and equation 6 respectively. Total runtime is ($< 24$ hours). Both LP and MILP estimation is ($< 30$ minutes) for each repetition. Please refer to table under A.5 for corresponding model URLS.

#### A.3.4 RISK STRATIFICATION EXPERIMENTS: SECTION 6.4

The dataset is available from Garcıa et al. (2024) upon reasonable request. Data is randomly split into calibration, estimation, training, and test sets. The models ECG-DL (Xiao et al., 2022), GBDT (Malinin et al., 2021) and FasterRisk (Liu et al., 2022) with the corresponding Github repos listed in the papers. The hyper parameters for GBDT are listed in Garcıa et al. (2024). The hyper-parameters for FR are listed in Garcıa et al. (2024). Once each model is trained, we apply class-conditional conformal (Lei, 2014) and our proposed Algorithm 1 on a Intel(R) Xeon(R) CPU E5-2683 v4 with 32Gb of memory and 125Gb of Disk. We use Gurobi to solve the proposed LP and MILP, that is formulations equation 7 and equation 6 respectively). We repeat this experiment ten times. Total runtime is ($< 10$ hours). Both LP and MILP estimation is ($< 15$ minutes) for each repetition.

#### A.3.5 EWMV EMPIRICAL UPPER BOUND: SECTION 6.7

To produce figure 6 we follow the ablation in figure 3.4 from Angelopoulos & Bates (2022). We fix $\alpha = .1$, and sample $R = 1000$ different estimation, calibration and test samples to produce the empirical CDF of EWMV's coverage. We repeat this for four different calibration sizes (500, 1000, 2500, 5000, 7500). We fix the estimation samples to 100. Aggregation is over three predictors (Swin, Vit, Resnet18), with the RAPS conformal method, for CIFAR-10 image classification.

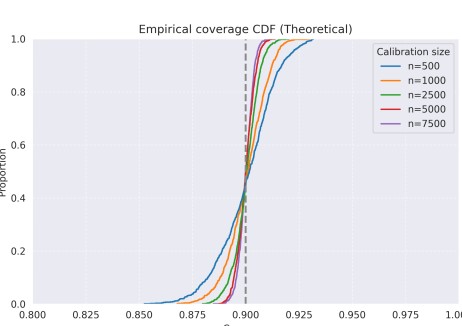

Figure 10: Empirical coverage estimate that nominal conformal predictors theoretically achieve for multiple calibration sample sizes and $(\alpha = 0.1)$.

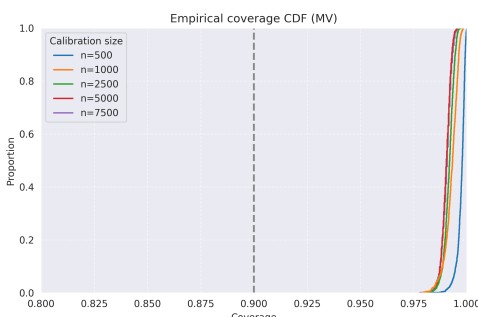

Figure 11: Empirical coverage estimate of MV across multiple calibration sample sizes $n_{\text{cal}}$ for $(\alpha = 0.1)$ using RAPS as the conformal method aggregating three CIFAR-10 predictors.

| Method | Inefficiency ($\downarrow$) | Validity ($\geq 0.90$) | Method | Inefficiency ($\downarrow$) | Validity ($\geq 0.99$) |
|---|---|---|---|---|---|
| k=10 | $5.745 \pm 0.186$ | $0.997 \pm 0.001$ | k=10 | $40.097 \pm 6.885$ | $1.000 \pm 0.000$ |
| k=9 | $3.233 \pm 0.118$ | $0.996 \pm 0.001$ | k=9 | $19.908 \pm 3.804$ | $1.000 \pm 0.000$ |
| Avg. | $3.089 \pm 0.089$ | $0.996 \pm 0.001$ | Avg. | $18.262 \pm 2.943$ | $1.000 \pm 0.000$ |
| YK | $2.525 \pm 1.079$ | $0.897 \pm 0.012$ | k=8 | $13.333 \pm 2.566$ | $1.000 \pm 0.000$ |
| YKadj | $2.377 \pm 1.548$ | $0.931 \pm 0.011$ | k=7 | $10.897 \pm 1.912$ | $1.000 \pm 0.000$ |
| k=8 | $2.351 \pm 0.083$ | $0.993 \pm 0.001$ | k=6 | $8.731 \pm 1.611$ | $0.999 \pm 0.000$ |
| YKsplit | $2.278 \pm 1.316$ | $0.898 \pm 0.016$ | k=1 | $7.849 \pm 2.873$ | $0.995 \pm 0.003$ |
| k=7 | $1.878 \pm 0.051$ | $0.990 \pm 0.001$ | Bonferroni | $7.849 \pm 2.873$ | $0.995 \pm 0.003$ |
| k=6 | $1.608 \pm 0.041$ | $0.986 \pm 0.002$ | MV | $7.603 \pm 1.777$ | $0.999 \pm 0.001$ |
| MD | $1.480 \pm 0.037$ | $0.985 \pm 0.002$ | k=5 | $7.602 \pm 1.747$ | $0.999 \pm 0.001$ |
| MV | $1.432 \pm 0.027$ | $0.981 \pm 0.003$ | k=4 | $6.316 \pm 1.681$ | $0.998 \pm 0.001$ |
| k=5 | $1.428 \pm 0.029$ | $0.981 \pm 0.002$ | k=3 | $6.209 \pm 2.266$ | $0.997 \pm 0.002$ |
| k=4 | $1.324 \pm 0.024$ | $0.975 \pm 0.004$ | MD | $6.077 \pm 1.750$ | $0.999 \pm 0.001$ |
| k=1 | $1.300 \pm 0.097$ | $0.947 \pm 0.009$ | k=2 | $5.968 \pm 2.219$ | $0.996 \pm 0.003$ |
| Bonferroni | $1.300 \pm 0.097$ | $0.947 \pm 0.009$ | YK | $3.662 \pm 0.487$ | $0.991 \pm 0.004$ |
| Single+RAPS | $1.299 \pm 0.051$ | $0.903 \pm 0.014$ | YKsplit | $3.601 \pm 0.736$ | $0.991 \pm 0.004$ |
| k=3 | $1.263 \pm 0.024$ | $0.971 \pm 0.004$ | YKadj | N/A | N/A |
| k=2 | $1.246 \pm 0.036$ | $0.962 \pm 0.005$ | Single+RAPS | $2.836 \pm 0.253$ | $0.991 \pm 0.003$ |
| **LP (Ours)** | $1.233 \pm 0.112$ | $0.962 \pm 0.042$ | **LP (Ours)** | $2.531 \pm 0.680$ | $0.993 \pm 0.005$ |
| **MILP (Ours)** | $1.123 \pm 0.045$ | $0.912 \pm 0.008$ | **MILP (Ours)** | $2.420 \pm 0.455$ | $0.992 \pm 0.003$ |
| Fisher* | $0.947 \pm 0.007$ | $0.886 \pm 0.007$ | Fisher* | $1.069 \pm 0.009$ | $0.944 \pm 0.005$ |

Table 6: CIFAR-100 baseline from section 6.6 repeated with error level $\alpha = 0.1$ and $\alpha = 0.01$. MILP and LP have the best performance. Fisher* is not valid. N/A means the adjusted error level $\bar{\alpha}$ from Yang & Kuchibhotla (2025) could not be computed for the current number of models and error level.

### A.3.6 BASELINE COMPARISON, EXTRA EXPERIMENTS

In table 6, we repeat the baseline comparison from section 6.6 with other error levels (i.e. $\alpha \in \{0.1, 0.01\}$). In table 7, we repeat the Imagenet experiment from 6.2 with the same baselines from table 6. To control computational complexity of p-value methods, we reduce the total sample size to 5000 and consider five models selected at random and $\alpha \in \{0.1, 0.05\}$. Results on both the CIFAR-100 and Imagenet experiments suggest both LP and MILP remain the most efficient of the methods that preserve validity.

### A.3.7 PERFORMANCE OF MULTIPLE SCORES AS A FUNTION OF THE ERROR LEVEL

We perform the CIFAR-10 experiment from section 6.2 with two different scores, APS Romano et al. (2020b) and $1 - p(y|x)$. We split the dataset into two-thirds for testing $D_{\text{test}}$ and one-third for

| name | Inefficiency($\downarrow$) | Validity($\geq 0.90$) | name | Inefficiency($\downarrow$) | Validity ($\geq 0.95$) |
|---|---|---|---|---|---|
| k=5 | $21.826 \pm 3.679$ | $0.994 \pm 0.001$ | k=5 | $54.636 \pm 13.819$ | $0.998 \pm 0.001$ |
| Avg. | $12.388 \pm 1.082$ | $0.993 \pm 0.003$ | Avg. | $25.395 \pm 3.146$ | $0.998 \pm 0.001$ |
| k=4 | $6.487 \pm 1.495$ | $0.988 \pm 0.004$ | k=4 | $14.338 \pm 1.956$ | $0.995 \pm 0.001$ |
| YKadj | $5.495 \pm 1.541$ | $0.940 \pm 0.030$ | YKadj | $8.935 \pm 12.023$ | $0.990 \pm 0.011$ |
| MD | $3.819 \pm 0.938$ | $0.984 \pm 0.006$ | MV | $7.237 \pm 0.616$ | $0.990 \pm 0.003$ |
| MV | $3.790 \pm 0.888$ | $0.984 \pm 0.006$ | YK | $6.615 \pm 0.778$ | $0.948 \pm 0.011$ |
| YK | $3.482 \pm 0.972$ | $0.898 \pm 0.041$ | YKsplit | $6.615 \pm 0.778$ | $0.948 \pm 0.011$ |
| YKsplit | $3.482 \pm 0.972$ | $0.898 \pm 0.041$ | k=3 | $6.287 \pm 0.325$ | $0.989 \pm 0.003$ |
| k=3 | $3.217 \pm 0.726$ | $0.980 \pm 0.008$ | MD | $3.881 \pm 0.353$ | $0.987 \pm 0.004$ |
| k=2 | $2.100 \pm 0.293$ | $0.965 \pm 0.012$ | Bonferroni | $3.485 \pm 1.595$ | $0.970 \pm 0.017$ |
| Single+RAPS | $1.787 \pm 0.047$ | $0.904 \pm 0.005$ | k=1 | $3.485 \pm 1.595$ | $0.970 \pm 0.017$ |
| Bonferroni | $1.759 \pm 0.260$ | $0.931 \pm 0.019$ | k=2 | $3.306 \pm 0.389$ | $0.979 \pm 0.005$ |
| k=1 | $1.759 \pm 0.260$ | $0.931 \pm 0.019$ | Single+RAPS | $2.595 \pm 0.215$ | $0.951 \pm 0.012$ |
| **LP (Ours)** | $1.590 \pm 0.159$ | $0.915 \pm 0.019$ | **LP (Ours)** | $1.964 \pm 0.189$ | $0.958 \pm 0.012$ |
| **MILP (Ours)** | $1.590 \pm 0.159$ | $0.915 \pm 0.019$ | **MILP (Ours)** | $1.964 \pm 0.189$ | $0.958 \pm 0.012$ |
| Fisher* | $1.155 \pm 0.044$ | $0.889 \pm 0.018$ | Fisher* | $1.296 \pm 0.033$ | $0.918 \pm 0.013$ |

Table 7: Baseline from section 6.6 repeated with error level $\alpha = 0.1$ and $\alpha = 0.05$ for the Imagenet task. MILP and LP have the best performance. Fisher* is not valid. N/A means the adjusted error level $\bar{\alpha}$ from Yang & Kuchibhotla (2025) could not be computed for the current number of models and error level.

calibration $D_n$. To evaluate methods (i.e. LP and MILP), we further split the calibration dataset $D_n$ into 70% for calibration $D_{cal}$ and 30% for estimation $D_{est}$. We also consider the model selection approaches (i.e. YK, YKsplit) from section 6.8 and the single smallest individual predictor on the test data (Oracle single model). Results in figure 12 and figure 13 suggest EWMV is more favorable w.r.t the smallest oracle model the smaller $\alpha$ is. We speculate this is because the factor of 2 correction in the error level affects less.

## A.4 EXTRAS

### A.4.1 LIST OF ACRONYMS

**APS**: Adaptive prediction sets
**RAPS**: Random adaptive prediction sets
**TRAQ**: Trustworthy retrieval augmented question answering
**LP**: Linear program formulation of EWMV
**MILP**: Mixed integer linear program formulation of EWMV.
**WMV**: Weighted majority vote.

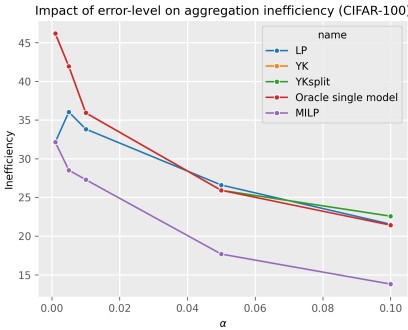

Figure 12: Inefficiency as a function of the error level for the APS score

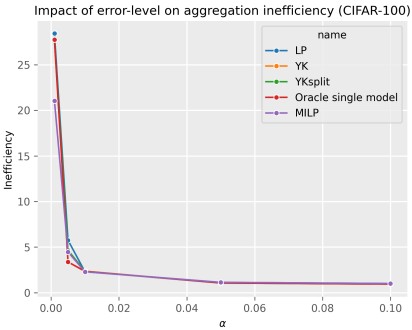

Figure 13: Inefficiency as a function of the error level for the $1 - p(y|x)$ score

**MV**: Majority vote.

## A.5    MODEL AND DATASET LIST

To aid reproducibility, we list the model and URL and license below grouped by each table under Section 6. We further list the datasets and conformal methods used.

| Model | URL |
|---|---|
| Alexnet | https://docs.pytorch.org/vision/main/models.html |
| Squeezenet | https://docs.pytorch.org/vision/main/models.html |
| MobileNet | https://huggingface.co/shehan97/mobilevitv2-1.0-imagenet1k-256 |
| Resnet50 | https://pytorch.org/hub/nvidia_deeplearningexamples_resnet50/ |
| Inception | https://docs.pytorch.org/vision/main/models.html |
| VGG19 | https://docs.pytorch.org/vision/main/models.html |
| ConvNext-large | https://docs.pytorch.org/vision/main/models.html |
| Wide-resnet101-2 | https://docs.pytorch.org/vision/main/models.html |
| Densenet161 | https://docs.pytorch.org/vision/main/models.html |
| Swin-b | https://docs.pytorch.org/vision/main/models.html |
| Regnet-Y-32GF | https://docs.pytorch.org/vision/main/models.html |
| Dinov2 | https://huggingface.co/facebook/dinov2-large-imagenet1k-1-layer |
| Vit-h-14 | https://docs.pytorch.org/vision/main/models.html |

| Model | URL |
|---|---|
| Resnet50 | jialicheng (a) |
| Swin-tiny-p4 | jaycamper |
| ConvNext | https://huggingface.co/karan99300/ConvNext-finetuned-CIFAR100 |
| Swin-tiny | MazenAmria (c) |
| Swin-small | MazenAmria (b) |
| Vit-base | jialicheng (b) |
| Vit-large | jialicheng (c) |
| Vit | Hugginface (2022) |
| Vit-base-in21k | pkr7098 |
| Swin-base | MazenAmria (a) |

| Model | URL |
|---|---|
| Resnet18 | https://huggingface.co/edadaltocg/resnet18_cifar10 |
| Swin | https://huggingface.co/Weili/swin-base-patch4-window7-224-in22k-finetuned-cifar10 |
| Vit | https://huggingface.co/MF21377197/vit-small-patch16-224-finetuned-Cifar10 |

| Model | URL |
|---|---|
| MiniLM | https://huggingface.co/deepset/minilm-uncased-squad2 |
| DynamicBert | https://huggingface.co/Intel/dynamic_tinybert |
| Roberta | https://huggingface.co/deepset/roberta-base-squad2 |
| DistillBert | https://huggingface.co/distilbert/distilbert-base-uncased-distilled-squad |
| MobileBert | https://huggingface.co/csarron/mobilebert-uncased-squad-v2 |

| Dataset | URL |
|---|---|
| Synthetic | Attached |
| Imagenet | https://huggingface.co/datasets/mlx-vision/imagenet-1k |
| Cifar-10 | https://huggingface.co/datasets/renumics/cifar10-outlier |
| Cifar-100 | https://docs.pytorch.org/vision/main/generated/torchvision.datasets.CIFAR100.html |

| Conformal method code | URL |
|---|---|
| TRAQ | https://github.com/shuoli90/TRAQ |
| RAPS | https://github.com/aangelopoulos/conformal-prediction |
| APS | https://github.com/aangelopoulos/conformal-prediction |
| Class-Conditional | https://github.com/jjgarciac/cc-risk-stratification |

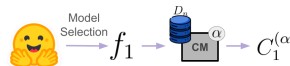

Figure 14: Standard **CP** pipeline to produce a $\alpha$-valid set-valued predictor $C_1^{(\alpha)}$ : $\mathcal{X} \to 2^{\mathcal{Y}}$ with calibration data $D_n$.

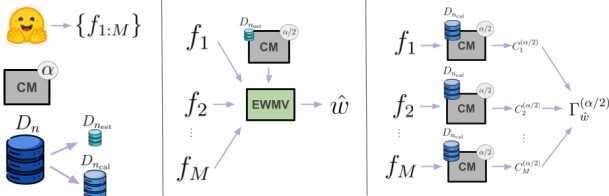

Figure 15: Proposed pipeline to produce a $\alpha$-valid set-valued predictor $\Gamma_{\hat{w}}^{(\alpha/2)}$ : $\mathcal{X} \to 2^{\mathcal{Y}}$ from calibration data $D_{\text{cal}}$ and estimation data $D_{\text{est}}$ where $D_n = D_{\text{cal}} \uplus D_{\text{est}}$. Note $\Gamma_{\hat{w}}^{(\alpha/2)}$ is a set-generating function and not a prediction set for a given input.

