# OpenReview forum: "EWMV: An algorithm to improve the efficiency of conformal methods"
_ICLR.cc/2026/Conference — Submitted to ICLR 2026_

### Official Review · Reviewer_nCBc · 2025-10-14

**Soundness:** 2
**Presentation:** 2
**Contribution:** 2
**Rating:** 2
**Confidence:** 5

**Summary:**

This submission proposes to minimize the average size of a CP-based prediction set over the weights used to combine different CP prediction sets. The approach is based on minimizing an empirical estimate of the average set size that is based in held-out data. Computationally, optimization is done either addressing a brute-force mixed integer linear program (MILP) formulation or using a standard linear program (LP) approximation obtained by replacing the indicator function with a hinge function.

A theoretical result is provided that demonstrates a sufficient condition under which the optimal weights do not increase the set size.

Experimental results are given for various tasks.

**Strengths:**

While the contribution is technically minor (see below), the submission presents some potentially interesting results on the benefits of CP set combination.

The paper is relatively clear, although the quality of the text could be significantly improved.

**Weaknesses:**

The proposed approach appears conceptually straightforward. The authors reserve a portion of the data and then optimize the prediction set size using this held-out subset. While this specific strategy may not have been explored in prior work, it does not seem to offer any formal guarantees. In particular, the main result concerning the expected set size (Proposition 5.2) states that if the optimal predictor’s prediction set size remains unchanged when increasing the coverage requirement, then optimizing the weight selection cannot reduce the average size. This conclusion is rather self-evident and does not directly address the challenges inherent in finite-sample, held-out data settings. Moreover, the paper does not clarify under what practical conditions the assumed invariance property of the optimal predictor would hold.

A further concern relates to data efficiency. Would it not be preferable to use the held-out data directly for calibration instead? In this context, one might have expected the authors to explore a meta-learning approach for weight optimization, allowing the procedure to rely solely on data from other tasks and thus avoid the need for task-specific held-out data.

**Questions:**

1. Can performance guarantees be provided that account for the use of finite held-out data and that do not require the strong assumption in Proposition 5.2?

2. Can the method be generalized to apply meta-learning across multiple tasks to avoid the need for held-out data?

---

> ### Author Response · Authors · 2025-12-03
>
> We appreciate the reviewer sees the potential of conformal model aggregation.
>
> In regards to the weaknesses, we split weakness 1 into multiple parts and clarify its components below:
>
> > W1.1: reserve a portion of the data and then optimize the prediction set size using this held-out subset.
>
> This is missing two non-trivial details. The error level needs to be updated to guarantee coverage and the optimization is framed in a tractable way. The latter is what enables EWMV to provide the efficiency gains of pre-conformalization approaches without their computational intractability.
>
> > W1.2 : While this specific strategy may not have been explored in prior work, it does not seem to offer any formal guarantees
>
> This is factually incorrect. Proposition 5.1 guarantees coverage.
>
> > W1.3: In particular, the main result concerning the expected set size (Proposition 5.2) states that if the optimal predictor’s prediction set size remains unchanged when increasing the coverage requirement, then optimizing the weight selection cannot reduce the average size. This conclusion is rather self-evident and does not directly address the challenges inherent in finite-sample, held-out data settings.
>
> This is a misunderstanding. Proposition 5.2 not the main result of the paper. The main proposal is the EWMV algorithm. Please see general point 1 and 2.
>
> > W1.4: Moreover, the paper does not clarify under what practical conditions the assumed invariance property of the optimal predictor would hold.
>
> We agree. To clarify, it can materialize when the distribution of the non-conformity score is discrete. An example setting is risk stratification, where the risk scores are sparse linear models with integer coefficients. These scores are used for medical triage and recidivism estimation. Please see [1] for more details. We added this remark to section 5.
>
> > W2: A further concern relates to data efficiency. Would it not be preferable to use the held-out data directly for calibration instead? In this context, one might have expected the authors to explore a meta-learning approach for weight optimization, allowing the procedure to rely solely on data from other tasks and thus avoid the need for task-specific held-out data.
>
> Surprisingly, not always the case. This is the main point we argue through-out the paper (Tables 1-5 and Figure 7). Across all our experiments every individual conformal predictor is computed with the entire calibration set and the corresponding user-specified error level ($\alpha$). EWMV only calibrates with the non-estimation subset of the data and with a more conservative error level ($\alpha/2$) to guarantee coverage. Meta-learning as an expected solution is speculative and is not grounded in the conformal model aggregation literature.
>
> In regards to the questions:
>
> > Q1: Can performance guarantees be provided that account for the use of finite held-out data and that do not require the strong assumption in Proposition 5.2?
>
> We conjecture not without stronger assumptions on the dependency of the conformal predictors. The root of the challenge stems from the validity constraint and is analogous to the discussion in section 6.7.
>
> > Q2: Can the method be generalized to apply meta-learning across multiple tasks to avoid the need for held-out data?
>
> This question is beyond the scope of the work.  Meta-learning as a way mitigate the need for held-out data for conformal model aggregation is speculative. To avoid/mitigate the need for held-out data, we'd recommend expanding the work of [2] (See section A.2.2).
>
> References:
>
> [1] Garcia, J.J., Sarin, N., Kitzmiller, R.R., Krishnamurthy, A.; Zègre-Hemsey, J.K.. (2024). Risk stratification through class-conditional conformal estimation: A strategy that improves the rule-out performance of MACE in the prehospital setting. Proceedings of the 9th Machine Learning for Healthcare Conference
>
> [2] Zeng, Hao, Kangdao Liu, Bingyi Jing, and Hongxin Wei. 2025. “Parametric Scaling Law of Tuning Bias in Conformal Prediction.” Paper presented at International Conference on Machine Learning.

---

### Official Review · Reviewer_TJCA · 2025-10-21

**Soundness:** 3
**Presentation:** 4
**Contribution:** 2
**Rating:** 4
**Confidence:** 4

**Summary:**

**Summary**

Conformal prediction provides a prediction sets that contain the true label with a guaranteed probability, but these sets are often inefficiently large, limiting their practical use.
This paper introduces EWMV (Estimating Weighted Majority Vote), an algorithm designed to address this inefficiency.
EWMV aggregates multiple conformal predictors into a single, more compact one.
It leverages the Weighted Majority Vote (WMV) framework, where a label is included in the final set if the sum of weights from predictors containing that label surpasses a threshold.
The core innovation of EWMV is its data-driven method for finding optimal weights. It uses a portion of the calibration data (an "estimation set") to learn weights that explicitly minimize the average prediction set size.
This learning problem is formulated as an optimization task, solvable via Mixed-Integer Linear Programming (MILP) for exactness or a faster
By constructing the base predictors at a more conservative error level (e.g., $\alpha/2$), the final aggregated predictor is guaranteed to maintain the desired coverage of at least $1-\alpha$.

**Contribution**

The paper's main contribution is an algorithm that improves the efficiency of conformal methods without sacrificing their core validity guarantees.
It provides a practical solution to the critical problem of selecting weights in the WMV framework, which was previously a major obstacle.
This work establishes a new paradigm for enhancing conformal prediction: instead of just selecting the best single model, it demonstrates that aggregating multiple readily available pre-trained models can yield superior results.
The authors provide theoretical justification for the algorithm's validity and present a sufficient condition for its efficiency gain over the best individual predictor.

**Strengths:**

**Originality**
The paper introduces EWMV, an algorithm for aggregating conformal predictors. Its primary originality lies in proposing a data-driven approach to learn the aggregation weights for the Weighted Majority Vote (WMV) framework. This addresses a key limitation of prior work where weight selection was often heuristic or suboptimal. By formulating the weight estimation as an empirical risk minimization problem solvable with standard optimizers (MILP or LP), the work provides a practical and principled method for improving the efficiency of prediction sets.

**Quality**
The work is well-grounded in the existing literature on conformal model aggregation. It clearly contextualizes its contribution by discussing and differentiating from prior p-value combination methods and set-based approaches. The paper builds upon the theoretical foundations of WMV to ensure the validity of the proposed aggregation scheme. This thorough background demonstrates a clear understanding of the field and helps to precisely situate the paper's novel contribution, strengthening the overall quality of the research.

**Significance**
The empirical evaluation is a significant strength of the paper. The authors validate their algorithm across a diverse set of four tasks, including image classification, question answering, and risk stratification, using multiple underlying conformal methods. This extensive testing demonstrates the general applicability and robustness of EWMV. The results consistently show that the proposed method improves efficiency over individual predictors and simpler aggregation baselines, providing strong evidence for the algorithm's practical utility.

**Weaknesses:**

The paper's experimental comparison could be strengthened by including other recent and concurrent work on the aggregation and selection of conformal prediction sets. Methods from Ge et al. (2024), Liang et al. (2025), Luo and Zhou (2024), Qin et al. (2024), and Yang and Kuchibhotla (2024) all address related problems. Comparing EWMV against these approaches would provide a more complete picture of its performance and position within the current literature, clarifying its relative advantages in efficiency and applicability.

A more detailed discussion of the methodological gap with score-level aggregation methods would be beneficial. For instance, Luo and Zhou (2024) also propose learning weights, but they aggregate the non-conformity *scores* before computing the quantile. In contrast, EWMV aggregates the final prediction *sets* after the quantile step for each predictor. Clarifying this distinction—aggregating pre- vs. post-conformalization—would help readers better understand the unique contribution and specific use case of the EWMV approach.

The weight estimation procedure on the estimation set $D_{nest}$ can be viewed as a form of tuning. As explored by Zeng et al. (2025) in other contexts, such data-dependent tuning can introduce a "tuning bias." The weights learned on a finite $D_{nest}$ may be slightly overfit, potentially leading to sub-optimal efficiency on unseen test data. The paper would be more complete if it acknowledged and briefly discussed this potential source of bias and how it might affect the generalization performance of the aggregated predictor.

#### References
Ge, Jiawei, Debarghya Mukherjee, and Jianqing Fan. 2024. “Optimal Aggregation of Prediction Intervals under Unsupervised Domain Shift.” 37 (November). https://openreview.net/forum?id=ldXyNSvXEr.

Liang, Ruiting, Wanrong Zhu, and Rina Foygel Barber. 2025. “Conformal Prediction after Data-Dependent Model Selection.” arXiv:2408.07066. Preprint, arXiv, July 3. https://doi.org/10.48550/arXiv.2408.07066.

Luo, Rui, and Zhixin Zhou. 2024. “Weighted Aggregation of Conformity Scores for Classification.” arXiv:2407.10230. Preprint, arXiv, July 14. https://doi.org/10.48550/arXiv.2407.10230.

Qin, Shenghao, Jianliang He, Bowen Gang, and Yin Xia. 2024. “SAT: Data-Light Uncertainty Set Merging via Synthetics, Aggregation, and Test Inversion.” arXiv:2410.12201. Preprint, arXiv, October 16. https://doi.org/10.48550/arXiv.2410.12201.

Zeng, Hao, Kangdao Liu, Bingyi Jing, and Hongxin Wei. 2025. “Parametric Scaling Law of Tuning Bias in Conformal Prediction.” Paper presented at International Conference on Machine Learning. Forty-Second International Conference on Machine Learning, June 18. https://openreview.net/forum?id=jnJLZXSOin.

**Questions:**

1. The experimental evaluation could be strengthened by including more recent baselines. Could you comment on how EWMV compares to other conformal aggregation and selection methods, such as those proposed by Yang & Kuchibhotla (2024) or Luo & Zhou (2024), and consider adding them to the empirical study?

2. The weight estimation on the set $D_{nest}$ is a form of data-dependent tuning. Could you discuss the potential for "tuning bias" in this step if $D_{nest}$ = $D_{cal}$? Specifically, how might the finite size of the estimation set affect the generalization of the learned weights and the efficiency of the final aggregated predictor on unseen data?

3. Could you make difference between EWMV and score-level aggregation approaches like that of Luo & Zhou (2024)? What are the key trade-offs between aggregating the final prediction sets (post-conformalization) versus aggregating the non-conformity scores (pre-conformalization)?

4. The paper focuses on classification. Have you considered extending the EWMV framework to regression problems for aggregating prediction intervals (such as Ge et al. (2024))? What would be the primary challenges in adapting the weighted majority vote mechanism and the optimization objective to handle continuous outputs?

I believe that responses to some or all of these points would significantly strengthen the paper, and I would be willing to increase my rating accordingly.

---

> ### Author Response · Authors · 2025-12-03
>
> We appreciate the author recognizes the originiality of the work, the groundness on existing literature and the extensive experiments to demonstrate the robustness of EWMV.
>
> In regards to weaknesses:
>
> > W1: The paper's experimental comparison could be strengthened by including other recent and concurrent work on the aggregation and selection of conformal prediction sets
>
> We agree. We include the "oracle" p-value aggregation methods explored in Qin et al. (2024) and the set aggregation approach of Yang and Kuchibhotla (2024) (See table 5 and figure 7). The approach of Ge et al. (2024) pertains to a different context (covariate shift). We could not add Luo \& Zhuo (2024) to our baseline comparison because it does not scale to the number of models we aggregate.
>
> > W2: A more detailed discussion of the methodological gap with score-level aggregation methods would be beneficial
>
> We agree. Methodologically, score level aggregation is limited by their scalability due to the dependency between weights and the conformalization (i.e. quantile estimation). For example, Luo and Zhou (2024) perform exhaustive search to estimate aggregation weights due to the conformalization being weight dependent. On the other hand, our proposal estimates aggregation weights post-conformalization, which supports two tractable optimization formulations (i.e. LP, MILP). Lastly, post-conformalization approaches are complimentary to pre-conformalization approaches and could be done to conformalize the sets that are then aggregated with our proposal. We added this dicussion to the related works.
>
> > W3: The paper would be more complete if it acknowledged and briefly discussed this potential source of bias and how it might affect the generalization performance of the aggregated predictor.
>
> We agree. Tunning bias could occur from using the calibration data for both estimation and calibration. We empirically explore this with the coverage gap metric proposed by Zeng et al (2025) of EWMV on the CIFAR-100 dataset. Results in figures 8 and 9 (Appendix A.2.2) for the LP and MILP formulations respectively suggest minimal difference between doing EWMV with only estimation data (hold-out) and doing EWMV with both calibration and estimation data (same). This matches results observed in figure 4 by Zeng et al (2025) and gives preliminary evidence to explore this as a way of mitigating the size of an estimation split.
>
> In regards to questions:
>
> > Q1: Could you comment on how EWMV compares to other conformal aggregation and selection methods, such as those proposed by Yang & Kuchibhotla (2024) or Luo & Zhou (2024), and consider adding them to the empirical study?
>
> Our work mitigates a tradeoff between these two proposals. The proposal of (Luo \& Zhou, 2024) improves efficiency beyond the single most efficient predictor but it does not scale because it  relies on exhaustive search over the space of aggregation weights. The proposal of (Yang \& Kuchibhotla, 2024) is computationally tractable but does model selection, and as such it does not improve efficiency beyond the single most efficient predictor. We show EWMV improves efficiency  beyond the single most efficient predictor (tables 1-5 and figure 7) and is computationally tractable (See section 6.5). We also added the work of (Yang \& Kuchibhotla, 2024) to the baselines in section 6.6 and section 6.8. EWMV report the most efficient sets among the valid approaches for both the LP and MILP formulations. We could not extend the baseline with (Luo \& Zhou, 2024) because the approach does not scale to the number of models we consider.
>
> > Q2: Could you discuss the potential for "tuning bias" in this step if the estimation and calibration datasets are the same?
>
> This is discussed in the response to W3.
>
> > Q3: Could you make difference between EWMV and score-level aggregation approaches like that of Luo & Zhou (2024)? What are the key trade-offs between aggregating the final prediction sets (post-conformalization) versus aggregating the non-conformity scores (pre-conformalization)?
>
> Aggregating post-conformalization considers a smaller subset of the label space, whereas aggregating pre-conformalization considers the entire label space. This becomes problematic for tasks with unwieldy label spaces (e.g. natural question answering in section 6.3). In terms of optimizing for efficiency, aggregating post-conformalization makes the optimization more tractable. Accordingly, our set level aggregation proposal scales better than the score level aggregation proposal of Luo \& Zhou (2024) without the need for hyperparameters. This is useful in situations in which we have more than three models available to aggregate (e.g. Section 6.2 or 6.3).

---

> > ### Author Response · Authors · 2025-12-03
> >
> > > Q4: Have you considered extending the EWMV framework to regression problems for aggregating prediction intervals (such as Ge et al. (2024))? What would be the primary challenges in adapting the weighted majority vote mechanism and the optimization objective to handle continuous outputs?
> >
> > We have considered extending EWMV to the regression case. The primary challenge stems from estimating the lesbegue measure of the aggregated set in a way that is tractable from an optimization perspective. The plausible formulations we have arrived to are different enough from EWMV that it warrants its own study. Thank you for the reference, we will consider it in the investigation of this case.

---

### Official Review · Reviewer_ZxS2 · 2025-10-30

**Soundness:** 3
**Presentation:** 3
**Contribution:** 3
**Rating:** 4
**Confidence:** 4

**Summary:**

This paper introduces EWMV (Efficient Weighted Majority Vote), an algorithm designed to aggregate multiple conformal predictors into a single, more efficient predictor that produces smaller prediction sets while preserving validity. Conformal prediction provides uncertainty quantification by outputting prediction sets that contain the true label with a guaranteed probability ($1-\alpha$). The authors leverage calibration data to estimate optimal weights for a weighted majority vote aggregation via linear programming (LP) or mixed-integer linear programming (MILP) formulations. Theoretical guarantees include validity preservation and a sufficient condition for efficiency improvement. Empirical evaluations across synthetic multiclass classification, image classification, natural question answering, and risk stratification demonstrate that EWMV often yields smaller sets than individual predictors or baselines like majority vote.

**Strengths:**

The paper presents new algorithms for weights computation for CP based on majority voting. Experimental results are extensive, demonstrating the performance on diverse tasks. Computational complexity seems reasonable.

**Weaknesses:**

1. The novelty of the proposed method is somewhat limited as it strongly relies on previous works on weighted majority voting, with the main difference of proposing practical algorithms for weight optimization.

2. Theoretical statements are either grounded in prior work (as in Proposition 5.1) or depend on assumptions that may be excessively strong (as in Proposition 5.2).

3. The paper does not refer to another approach for conformal ensembling at the predictor level, via score weighting or multi-dimensional CP. See for example:
[1] Bai, Yu, et al. "Efficient and Differentiable Conformal Prediction with General Function Classes." International Conference on Learning Representations (2022). - see Appendix F
[2] Luo, Rui, and Zhixin Zhou. "Weighted aggregation of conformity scores for classification." arXiv preprint arXiv:2407.10230 (2024).‏
[3] Tawachi, Yam, and Bracha Laufer-Goldshtein. "Multi-dimensional conformal prediction." The Thirteenth International Conference on Learning Representations. 2025.‏

**Questions:**

Please address the issues outlined in the Weaknesses section, as well as the additional questions listed below:
1. Proof of proposition 5.2 - why is the first inequality valid? Eq. (5) is an optimization on an empirical estimate, while the proposition deals with the actual expectation. In addition, the assumption "if the average size of the most efficient predictor does not change when we re-estimate it at a more conservative level” does not seem to hold in practice as it almost always the case that the set size increases for more conservative levels, and especially when decreasing $\alpha$ by a factor of $2$.
2. How does the method work with other nonconformity scores, such as $1-p(y|x)$?
3. Comparisons to other baselines is only shown for CIFAR-100 with $\alpha=0.05$ (which is different from $\alpha=0.1$ used for the same experiment in Table 2. Could you provide additional comparisons  over other datasets/tasks and other coverage levels?
4. line 419: “For reference, we also plot the runtime of MD (From section 6.1)” - what is MD?
5. Figure 7 is unclear - why is MILP performance fixed across the number of models - can you show the performance as a function of the number of heads for both proposed methods (MILP and LP) as well as other baselines?
6. Minor - “We use RAPS from Angelopoulos et al. (2022) as the conformal method and obtain all the fine-tuned models along with the dataset are available from HuggingFace and Torchvision” - “are available” seems redundant

---

> ### Author Response · Authors · 2025-12-03
>
> We appreciate the reviewer considers the experimental results are extensive and the computational complexity is reasonable.
>
> In regards to the weaknesses
>
> > W1: The novelty of the proposed method is somewhat limited as it strongly relies on previous works on weighted majority voting, with the main difference of proposing practical algorithms for weight optimization.
>
> Please see general point 1.
>
> > W2: Theoretical statements are either grounded in prior work (as in Proposition 5.1) or depend on assumptions that may be excessively strong (as in Proposition 5.2).
>
> We agree proposition 5.1 is grounded in WMV. We also agree proposition 5.2 is strong. However, it can materialize in practice for discrete non-conformity scores (e.g. those that stem from risk scores) and where $\alpha$ is often small (Please see (Garcia et al., 2024) for more details on this point).
>
> > W3: Extra referenes
>
> Thank you for those references. We added them to the related works and characterize them as pre-conformalization aggregation approaches. Our work is complimentary to pre-conformalization aggregation in two ways: (1) Our optimization formulation is tractable, scales in the number of models and converges without the need for hyperparameters. This methodologically expands previous optimization formulations that rely on min-max formulations or exhaustive search. (2) Post-conformalization approaches can be done after pre-conformalization, for example we could aggregate multiple scores pre-conformalization and then aggregate multiple predictors post-conformalization. More details can be found in the related works.
>
> In regards to the questions:
>
> > Q1: Proof of proposition 5.2 - why is the first inequality valid? Eq. (5) is an optimization on an empirical estimate, while the proposition deals with the actual expectation. In addition, the assumption "if the average size of the most efficient predictor does not change when we re-estimate it at a more conservative level” does not seem to hold in practice as it almost always the case that the set size increases for more conservative levels, and especially when decreasing $\alpha$ by a factor of 2
>
> This is a typo. The first line should say inequality (3) and not (5). In regards to the assumption "if the most efficient predictor does not change when we re-estimate it at a more conservative level": while it is true that sizes often increase as $\alpha$ decreases, it is not always true for an important class of  predictors (i.e., risk scores), and for very small values of $\alpha$ (as is often required for risk score). Lastly, we emphasize the assumption is not a necessary for our approach to work. We provide ample empirical evidence EWMV improves efficiency over the smallest individual predictor at level $\alpha$ across various conformal methods and tasks.
>
> > Q2: How does the method work with other nonconformity scores?
>
> EWMV improves efficiency with other non-conformity scores. For example, APS score in figure 2, RAPS score in table 2, negative similarity between questions and passages (TRAQ) in table 3, and negative risk conditioned on outcome (CC) in table 4. We repeated the CIFAR-100 baseline with the APS score and the $1-p(y|x)$ score (Section A.3.7). Figures 12, 13 suggest EWMV outperform the other methods all error levels for ACS and small ones for $1-p(y|x)$.
>
> > Q3:  Could you provide additional comparisons over other datasets/tasks and other coverage levels for the baselines?
>
> Yes. We performed the same experiment on CIFAR-100 and Imagenet for various alpha values in Tables 5,6,7. Results suggest MILP/LP still produces the most efficient sets among the baselines. Other tasks like Open QA do not directly support pre-conformalization approaches like p-value aggregation. This highlights an advantage of aggregating post-conformalization.
>
> > Q4: line 419: “For reference, we also plot the runtime of MD (From section 6.1)” - what is MD?
>
> MD stands for Mirror Descent. It is an iterative optimization algorithm to solve the LP formulation of EWMV. Please find more information in Section 6.1.
>
> > Q5: why is MILP performance fixed across the number of models - can you show the performance as a function of the number of heads for both proposed methods (MILP and LP) as well as other baselines?
>
> The red line corresponded to the performance of MILP with all 13 models. We repeated the experiment and now plot the efficiency of MILP, LP, and the baselines in section 6.6,  as a function of the number of models. Results in Figure 7 suggest MILP and LP are the consistently most efficient of the valid approaches when aggregating two or more models.
>
> > Q6: Minor - “We use RAPS from Angelopoulos et al. (2022) as the conformal method and obtain all the fine-tuned models along with the dataset are available from HuggingFace and Torchvision” - “are available” seems redundant
>
> Thank you for catching that. We deleted the redundant statement.

---

### Official Review · Reviewer_3Gtr · 2025-11-10

**Soundness:** 3
**Presentation:** 2
**Contribution:** 2
**Rating:** 6
**Confidence:** 2

**Summary:**

The paper proposes an approach, EWMV, that aggregates the predictions of multiple conformal predictors into a single prediction set. The authors show through empirical investigation that EWMV usually generates more efficient prediction sets than the aggregated predictors. The evaluations also included a range of tasks and multiple conformal prediction methods.

**Strengths:**

- The paper is proposing a novel approach that improves the informational efficiency of predictions.
- The evaluation is thorough and has been conducted across multiple tasks.
- The results show efficiency enhancement with EWMV compared to the competing approaches.
- Overall, the paper can be followed and understood with relative ease.

**Weaknesses:**

- The paper provides an incremental improvement over the weighted majority voting algorithms. In other words, the novelty is limited.

- The computational complexity is a drawback of the proposed approach (exponential for MILP).

- Some acronyms were used in the text before being introduced.

- In section 6.7, coverage needs to be clearly defined.

- There is also a typo in the first sentence of the abstract.

**Questions:**

1- In what sense sections 6.6 and 6.7 provide ablation studies? i.e., what part of EWMV has been removed to study its effect on the performance? Section 6.6, for instance, looks like a classic comparison between different methods.

2- Line 426 starts with "In the table", which table?

---

> ### Author Response · Authors · 2025-12-03
>
> We appreciate the reviewer finds our proposal novel, the evaluation thorough, the efficiency improvements apparent and the paper clear.
>
> In response to weaknesses:
>
> > W1: The paper provides an incremental improvement over the weighted majority voting algorithms. In other words, the novelty is limited.
>
> Please see general point 1.
>
> > W2: The computational complexity is a drawback of the proposed approach (exponential for MILP).
>
> The worst case exponential complexity is not a drawback of EWMV, but rather of the MILP formulation. The LP formulation of EWMV has polynomial runtime complexity and is a valid alternative to MILP, as it also provides efficiency better than the individual most efficient predictor and other baselines (See tables 1-5 and figure 7). We'd also like to emphasize the exponential complexity is a worst case statement on the number of estimation samples and labels; across our experiments, this led to a run time of 12min (See figure 5).
>
> > W3: Some acronyms were used in the text before being introduced.
>
> We added a table with acronyms to appendix A.4.1
>
> > W4: In section 6.7, coverage needs to be clearly defined.
>
> We added the definition to section 6.7
>
> > W5: There is also a typo in the first sentence of the abstract.
>
> We did not find a typo in the first sentence of the abstract.
>
> In response to questions:
>
> > Q1: In what sense sections 6.6 and 6.7 provide ablation studies? i.e., what part of EWMV has been removed to study its effect on the performance? Section 6.6, for instance, looks like a classic comparison between different methods.
>
> Thank you for pointing out the confusion. We deleted the word "Ablation" and changed the section name to "additional experiments".
>
> > Q2: Line 426 starts with "In the table", which table?
>
> Table 5.

---

### Author Response · Authors · 2025-12-03
**General points**

We'd like to thank the reviewers for their insightful questions,  comments and overall engagement with the work. We are pleased to hear the paper was clear and that the experiments demonstrated the performance, potential and robustness of our proposal.

A couple general points we would like to clarify:

1. Our work builds upon the weighted majority vote (WMV) algorithm but it is not an incremental improvement over it. WMV characterizes a class of combined conformal predictors that are valid but not necessarily serviceable from an efficiency perspective (See Figure 1). The novelty of our approach lies in optimizing over this class of functions, with tweaks in error level and a held-out estimation dataset, to uncover, in a scalable way, a more efficient predictor than each individual component. This contributes, to the best of our knowledge, the most computationally tractable conformal aggregation algorithm that still manages to improves efficiency over each individual predictor validated on multiple tasks, conformal methods and aggregation baselines. This mitigates a tradeoff in the conformal model aggregation literature, where algorithms either improve efficiency over individual predictors but are computationally intractable (e.g. relying on exhaustive search or min-max formulations); or are computationally tractable but do not improve efficiency over all individual predictors.
2. Proposition 5.2 establishes a sufficient but not necessary condition to guarantee efficiency improvements. We provide ample empirical evidence EWMV improves efficiency without this assumption in Tables 1-5 and Figure 7. Due to this confusion, we have moved proposition 5.2 to the appendix, label it proposition A1 and mention it as a remark in the main body.

We also reply to each reviewer's comment separately.

---

### Meta-Review · Area_Chair_VNvL · 2026-01-16

**Summary:**

The key concerns raised by multiple reviewers center on (1) limited novelty relative to prior Weighted Majority Vote (WMV) conformal aggregation, (2) data inefficiency / tuning via an extra estimation split (which seems necessary here), and (3) a gap between empirical weight optimization and theoretical understanding of efficiency/generalization (But obtaining such a bound is arguably more tricky. Btw, I would suggest a look at this recent article https://arxiv.org/abs/2306.07254). While the paper clearly preserves validity by building base predictors at a more conservative error level and provides extensive empirical results, several reviewers argued the main conceptual step that is holding out data to learn weights that minimize empirical average set size, feels straightforward and may not justify the claimed contribution. I personally believe that simplicity is not a negative property and this should not be considered as critics.  However, the evaluation does not fully resolve the fair data-budget question: EWMV requires splitting calibration data into calibration and estimation subsets, whereas competing aggregation baselines often do not, and the paper does not provide an apples-to-apples comparison under a fixed calibration budget or a thorough efficiency study of a no-split variant. Finally, beyond coverage guarantees, reviewers noted the paper offers limited guarantees about how well the learned weights generalize in finite samples (this point is quite fundamental, because the additional learning step introduced can overfit. This is not deeply explored in the paper); Proposition 5.2 was viewed as relying on strong assumptions and was not convincing as a main efficiency guarantee. The rebuttal improves clarity and positioning (and de-emphasizes Prop 5.2), but it does not fully address the novelty perception and data-efficiency/generalization concerns, leading to an overall weak reject recommendation.

I enjoyed reading the paper and discussions but the overall conclusion would be to lean toward a weak reject.

**Reviewer Concerns:**

See comment above

**Reviewer Scores:**

Being optimistic, I would guess

- Reviewer 3Gtr: 6 to 6. Clarity issues were fixed, but their main “novelty is limited” concern likely remains. This is hard to change mind in short discussion period

- Reviewer ZxS2 (4): 4 to 5. Authors addressed most technical points (issues/positioning, added refs, improved Fig. and broader comparisons).

- Reviewer TJCA: 4 to 5. Better positioning vs related work + added tuning-bias discussion/evidence likely moves them slightly up.

- Reviewer nCBc (2): 2 to 2. Core objections (data-splitting inefficiency + lack of finite-sample efficiency guarantees) aren’t fundamentally resolved. I would not see a score change here without significant additional material

---

### Decision · Program_Chairs · 2026-01-26

Reject